# Unusual Mass Mortality of Atlantic Puffins (*Fratercula arctica*) in the Canary Islands Associated with Adverse Weather Events

**DOI:** 10.3390/ani15091281

**Published:** 2025-04-30

**Authors:** Cristian M. Suárez-Santana, Lucía Marrero-Ponce, Óscar Quesada-Canales, Ana Colom-Rivero, Román Pino-Vera, Miguel A. Cabrera-Pérez, Jordi Miquel, Ayose Melián-Melián, Pilar Foronda, Candela Rivero-Herrera, Lucía Caballero-Hernández, Alicia Velázquez-Wallraf, Antonio Fernandez

**Affiliations:** 1Unit of Veterinary Histology and Pathology, University Institute of Animal Health and Food Safety (IUSA), Veterinary School, University of Las Palmas de Gran Canaria (ULPGC), 35413 Las Palmas de Gran Canaria, Canary Islands, Spain; cristian.suarez@ulpgc.es (C.M.S.-S.); oscar.quesada@ulpgc.es (Ó.Q.-C.); candela.rivero101@alu.ulpgc.es (C.R.-H.);; 2Departamento de Obstetricia y Ginecología, Pediatría, Medicina Preventiva y Salud Pública, Toxicología, Medicina Legal y Forense y Parasitología, Universidad de La Laguna, Av. Astrofísico F. Sánchez, sn, 38203 La Laguna, Canary Islands, Spain; rpinover@ull.edu.es (R.P.-V.); jordimiquel@ub.edu (J.M.);; 3Instituto Universitario de Enfermedades Tropicales y Salud Pública de Canarias, Universidad de La Laguna, Av. Astrofísico F. Sánchez, sn, 38203 La Laguna, Canary Islands, Spain; 4Servicio de Biodiversidad, Dirección General de Lucha Contra el Cambio Climático y Medio Ambiente, Gobierno de Canarias, Cl. Profesor Agustín Millares Carló, 18. 5ª Planta, Edificio Servicios Múltiples II, 35071 Las Palmas, Canary Islands, Spain; 5Gestión y Planeamiento Territorial y Medioambiental, S.A. (GESPLAN), Canary Islands Government, C/León y Castillo 54, Bajo, 35003 Las Palmas de Gran Canaria, Canary Islands, Spain

**Keywords:** Atlantic puffin, *Fratercula arctica*, *Renicola sloanei*, mass mortality, starvation, nephropathy, migration, weather adverse event

## Abstract

The Atlantic puffin (*Fratercula arctica*) is a seabird species characterized by great diving capabilities and transoceanic migratory behavior. An unusual mortality event of Atlantic puffins was detected from late 2022 to early 2023 along the coast of the Canary Islands. This unusually high mortality raised social, health, and environmental concerns related to a possible infectious outbreak. Diagnostic laboratory analyses were performed on dead Atlantic puffins. The finding demonstrated that the birds were very thin and weak because of extreme loss of muscle and fat, indicating malnutrition. A renal parasite, identified as *Renicola sloanei,* was observed in a significant proportion of individuals. The investigation performed in relation to this mortality event of Atlantic puffins concluded that starvation and adverse climatic events during migration should be considered as risks factors contributing to mass mortality in this species.

## 1. Introduction

The Atlantic puffin (*Fratercula arctica*) is a seabird renowned for its exceptional diving capabilities and transoceanic migratory behavior. Phylogenetic analyses have identified several subspecies that nest in various geographical locations, including northern Europe, Greenland, and North America [1,2,3]. Its diet is rich in small fish, which may vary according to different subspecies, but frequently include sand eels (*Hyperoplus* spp.), herrings (*Clupea* sp.), and capelin (*Mallotus* sp.) [4].

Significant population declines of European Atlantic puffins have been linked to reduced recruitment and prey availability [2,4,5,6]. In Europe, juvenile puffins tend to migrate farther south than adults. These movements contribute to the dispersion of the species during migration, although episodes of mortality associated with migration may be a normal occurrence in the dynamics of Atlantic puffin populations [5,7]. However, there is limited literature on the pathological investigation of unusual mortality events in wild Atlantic puffins [2,8,9].

Digenean trematodes of the genus *Renicola* are relatively common parasites found in marine birds, including Atlantic puffins [10,11]. The adult fluke parasitizes the urinary system of seabirds, but the complex life cycle of the helminth requires several intermediate hosts before reaching the bird [11]. The metacercarial stage is the infective form for the seabird, and they are encysted in the mesenteries connecting the pyloric caeca of the fish, which would be acting as an intermediate host. The metacercarial can be commonly found in European sprat (*Clupea sprattus*) and mediterranean sardines (*Sardina pilchardus*). The seabird usually acquires the parasite through the ingestion of infected fish. After the excystation of the metacercaria in the bird gastrointestinal tract, the parasites migrate to reach their definitive location at the bird urinary tract [11]. The role of *Renicola* infestation in the development of renal lesions in Atlantic puffins remains unclear, as the presence of the trematode is frequent but not typically associated with disease according to previous reports [11,12].

A series of winter storms battered the coasts of the Iberian Peninsula, France, and the United Kingdom between December 2022 and January 2023. These storms were named by various meteorological organizations as storms Efraín, Fien, and Gérard. In particular, the formation of the storm Fien was rapidly followed by the storm Gérard. They were characterized by the existence of a very powerful jet stream (exceeding 300 km/h) at high atmospheric levels, which crossed the entire North Atlantic from the Labrador Peninsula to the Iberian Peninsula (see detailed meteorological information in https://www.aemet.es/es/conocermas/borrascas/2022-2023 [accessed on 20 April 2025]). Extreme weather conditions may represent a life-threat event, even for a well-adapted seabird. As global warming is accompanied by an increase in the frequency of extreme weather events, this may represent a new scenario for the Atlantic puffins that become lost from their migratory route.

This study presents the pathological findings of Atlantic puffins involved in an unusual mortality event observed from late 2022 to early 2023 along the coast of the Canary Islands. This event was temporally and geographically associated with a series of winter storms during the non-breading season.

## 2. Materials and Methods

### 2.1. Stranding Records

Between 4 January and 14 February, 2023, a total of 223 Atlantic puffins (*Fratercula arctica*) were submitted to the University Institute of Animal Health and Food Security at the University of Las Palmas de Gran Canaria for investigation of the cause of death. The necropsies were performed in cooperation with the “Dirección General de Lucha Contra el Cambio Climático y Medio Ambiente,” as part of the Canarian Network for the Surveillance of Wildlife Health (Orden Nº134/2020 of 26 May 2020).

The animals were found at various points along the coast of the Canarian archipelago, coinciding with a series of winter storms that originated in the north Atlantic. In Figure 1, there is a satellite image of the Efraín storm, in which the wintering and breeding distribution of the Atlantic puffins have been represented. Carcasses were submitted by various entities, including local authorities, environmental police, and wildlife hospitals. The Canarian Network for the Surveillance of Wildlife Health collected the information of the stranding episodes among the islands, which guaranteed the traceability of the events. In the Appendix A, there is extra information about the locations and dates of the finding of the animals, including the UTM coordinates of each event.

### 2.2. Necropsy and Histopathology

The degree of decomposition was assessed upon reception, with a numeric value assigned between 1 and 5 (1—very fresh, 2—fresh, 3—decomposed, 4—very decomposed, 5—skeletal reduction). Complete necropsy following standardized procedures [13] was performed on animals with decomposition codes of 1–3, but a total of 142 decomposed animals were excluded. Body score (BS) was determined using a numeric scale from 1 to 3 (1—cachexia, 2—thin, 3—optimal) based on the evaluation of muscle mass and subcutaneous and visceral fat (a modification of the body score presented by Burton et al. [14]). Biological data were obtained from each individual, including total body weight, sex, and gonad development. The animals were categorized as mature or immature based on external features and gonad development [13]. When the feathers of the animal were covered by sand or wet, the weight was considered not representative and excluded from the analysis (*n* = 87).

Tissue samples for histological examination included adrenal glands, air sacs, bursa of Fabricius, encephalon, esophagus, eyes, gonads, heart, intestine, kidneys, liver, lungs, proventriculus, sciatic nerves, skeletal muscle, skin, spleen, thymus, thyroid glands, trachea, ureters, ventriculus, and the whole head. Samples were fixed in 4% neutral-buffered formalin for 24 h, routinely processed, and embedded in paraffin. Sections of 4 μm were stained with hematoxylin and eosin for histological analysis. Microscopic examination was conducted by two certified veterinary pathologists (O.Q.C. and A.F.).

### 2.3. Molecular Analysis

Molecular analyses were performed for the exclusion of avian influenza virus and parasitological identification. A simultaneous DNA/RNA extraction with the magnetic bead method using a robotic platform was performed, following the manufacturer’s protocol for the ZYMO DNA/RNA extraction kit (ZYMO Research). To ensure the accuracy and reliability of the extraction process, both a negative control (nuclease-free water) and a positive control, were included in the protocol.

Tissues from 45 animals were also tested for the presences of highly pathogenic avian influenza (H5N1) RNA using a real-time semi-quantitative (sq) PCR (polymerase chain reaction) [15].

Genomic DNA was extracted using preserved tissue portions. PCR screening of DNA was based on cytochrome *c* oxidase subunit 1 (*cox1*) using the primers CO1-R and JB3. The PCR products that presented the expected size (650-bp) were sequenced at Macrogen (Madrid, Spain) with primers CO1-R and JB3 [16].

## 3. Results

### 3.1. Stranding Records

Of the 223 Atlantic puffins included in our records, 67.7% (*n* = 151) of them came from the eastern islands, including Lanzarote (*n* = 68), Gran Canaria (*n* = 45), La Graciosa (*n* = 20), and Fuerteventura (*n* = 18). The remaining 32.3% of individuals (*n* = 72) were submitted from the western islands: La Palma (*n* = 19), Tenerife (*n* = 17), La Gomera (*n* = 15), and El Hierro (*n* = 21). The inset of Figure 1 represents the distribution of the strandings episodes along the coast of the Canary Islands, obtained from the geolocation of each wreck event.

### 3.2. Necropsy and Histopathology

The registered mean body mass of 136 individuals was 256.0 ± 47.4 g. The individual data can be found in the Appendix A. Complete standardized necropsy was performed on 81 animals. There were 54 females (66.66%) and 27 males (33.34%). Both juveniles (*n* = 52) and adults (*n* = 29) were present.

The most consistent gross findings were generalized muscle atrophy (Figure 2A) and serous fat atrophy (Figure 2B). These conditions were evident in 96.3% (78/81) of the individuals for which body score (BS) could be determined. The BS ranged from cachectic (BS1, *n* = 40) to thin (BS2, *n* = 38), with only 2.5% of individuals (*n* = 2) having a BS of 3. In all animals, the ventriculus and proventriculus were empty (Figure 2C), except for the frequent presence of microplastics (i.e., fragments of plastic less than 5 mm) in the ventriculus.

Thirteen animals exhibited lesions indicative of acute blunt-force trauma, including skin abrasions and lacerations, subcutaneous hematomas, internal hemorrhages, and bone fractures.

Histological analysis was performed on 81 animals, and the findings are summarized in Appendix A. The primary lesions were found in the kidneys of all the animals analyzed and included dilation and inflammation of the primary ureter branch and medullary cones, with the occasional presence of intraluminal helminths (Figure 3A,B). Marked epithelial hyperplasia was detected in 88.8% (72/81) of cases, with concurrent moderate-to-severe squamous metaplasia in 18.5% of individuals (*n* = 15) (Figure 3C). The collecting ducts of the medullary cone were dilated, containing abundant cellular debris. There was severe lymphoplasmacytic inflammation of the subepithelial connective tissue, with occasional heterophils and macrophages (inset of Figure 3C). While ureteritis was noted in 100% (*n* = 81) of the animals, it ranged from mild (*n* = 3) to severe (*n* = 54). Concomitant interstitial lymphoplasmacytic nephritis was observed in 59.3% (48/81) of cases. In 82.7% of animals (67/81), there was urate deposition in the form of crystals or spheres in the renal tubules (Figure 3D). Acute tubular necrosis was noted in 8.6% (7/81) of individuals (Figure 3E).

Adult trematodes or their eggs were detected in the collecting ducts or the primary branch of the ureter in 27.2% of individuals (22/81). The trematodes were observed to infect 22.2% of males (6/27) and 29.6% of females (16/54). Regarding bird maturity, 30.8% (16/52) of immature and 20.7% of mature (6/29) Atlantic puffins suffered from the trematode infection.

Additional relevant histological findings were noted in the skeletal muscle. The pectoral muscles of 30 individuals were evaluated, all of which showed severe diffuse myocyte atrophy. Acute to subacute lesions were observed in the pectoral muscles of 19 animals, including moderate to severe acute degeneration and segmental necrosis of myocytes, with the occasional presence of macrophages phagocytizing cellular debris.

### 3.3. Molecular Analysis

All the animals resulted negative for highly pathogenic avian influenza.

The recovered digeneans were identified as belonging to the genus Renicola, 1904 (Renicolidae) [10,16,17], with morphoanatomical characteristics resembling those of *Renicola sloanei* [11]. From the PCR, a fragment of 527 bp was obtained for the region of the mitochondrial cytochrome c oxidase subunit I gene. The BLAST analysis showed greater homology with Renicola sloanei (Accession Number: MK463857, Query Cover: 99%, Identity: 87.68%). The 527 bp nucleotide sequence obtained in this study was submitted to the GenBank database (accession number OQ992508). To see a detailed description of the parasite morphology and molecular identification, see Pino-Vera et al. [16].

## 4. Discussion

The Atlantic puffin is a rare visitor to the Canarian coast, as this archipelago lies at the margin of the wintering distribution of the species [18]. As part of the Canarian Network for the Surveillance of Wildlife Health, only four animals were submitted for necropsy between 2021 and 2022. The atypical stranding of hundreds of Atlantic puffins in January 2023 alerted the authorities and the public, prompting this investigation.

Burnham et al. [7] reported different migratory movements for males and females, with females migrating with more segregation and a more southern distribution than males. This difference in migratory patterns may explain why we observed more females than males involved in this mortality event.

There are several populations of Atlantic puffins that may show significant variation in body mass [1,3]. However, the reported weight for the species during the non-breeding season is between 390 and 730 g [2]. Variations can occur according to sex (males are slightly heavier), breeding period, or migration, but Atlantic puffins are expected to gain weight during winter [2,19]. The biometric results indicate that most of our cases were below the normal range, with a median body mass of 256.0 g. This is drastically lower than the reported healthy weight for different populations of wild Atlantic puffin [1,2,3]. This low body mass is indicative of a systemic alteration, such as malnutrition or illness.

Regarding illness, we excluded highly pathogenic avian influenza but confirmed a high prevalence of renal lesions associated with infestation by *Renicola sloanei* [16]. *R. sloanei* has been associated with mild lesions in different species of penguin (Spheniscidae), the common murre (*Uria aalge*), and the Manx shearwater (*Puffinus puffinus*) [10,20].

In previous studies in the United Kingdom, a high rate of infection was detected in Atlantic puffins, but the histopathology usually revealed an apparent lack of host–tissue reaction, dilation of ducts due to the presence of worms, and epithelial erosion [11].

To our knowledge, there is a single finding of an unidentified species of the genus *Renicola* in Britain [12] causing “nephrosis” in Atlantic puffins. Wright [11] also described a case of an exhausted Atlantic puffin found in the London Docks that died shortly after, with the cause of death determined as renal insufficiency. Approximately 40 flukes were found in the kidney, corresponding to *Renicola* sp. closely related to *Renicola sloanei*.

It would be expected that a well-adapted host–parasite relationship would result in minimal to mild lesions. In the literature reports of *R. sloanei*, the trematode is visualized inside the ureter with no associated inflammatory reaction [10,11]. In a pathological study of *Fratercula cirrhata*, renal trematodiasis was not found to be a cause of mortality [21].

Although histologically we detected the trematode in only 27.2% of cases, we observed moderate to severe lymphoplasmacytic inflammatory reactions in 96.3% of the puffins. This may suggest a much higher degree of infestation, not detected during routine histopathology, although other causes cannot be ruled out. The trematode is relatively small (around 1600 μm), and adults may be located anywhere in the urinary tract, including ureters or main ureteral branches of the different renal divisions [11]. Careful investigation of the ureters during gross examination, and the inclusion of representative samples of different renal divisions, as well as ureters for histopathology, may increase the chance of detecting the trematode. Although our histopathological analysis revealed some early data of parasitic incidence of *Renicola sloanei* in the Atlantic puffin, this diagnostic method may underestimate the actual infestation rate. Clarification of the best diagnostic tool for this condition merits further investigation; however, molecular analysis with the amplification of the NADH dehydrogenase subunit 1 gene may be more sensitive and specific for detecting the parasite in renal tissue.

Two important factors in exerting density-dependent effects are host immune responses and competitive interactions between trematodes. Both these factors may lead to host mortality as trematode density increases [22]. The nutritional status of birds can influence parasite dynamics and parasite virulence, with undernourished birds having more parasites, causing more tissue damage [23].

Vitamin A deficiency in avian species is a well-recognized syndrome causing squamous metaplasia of stratified epithelium in various body locations, including conjunctiva, lacrimal and salivary glands, esophageal glands, and respiratory and ureteral epithelium [24]. We observed moderate grades of squamous metaplasia of the ureteral epithelium in 18.5% of the analyzed animals. Chronic ureteral irritation caused by the presence of the trematode is one important predisposing cause for squamous metaplasia [20].

The presence of histological lesions, such as epithelial degeneration, necrosis, and visceral gout (urate deposition in the renal tubules) suggests a significant metabolic imbalance affecting the individual’s homeostasis. These lesions were observed in association with ureteral obstruction—caused by ureteral inflammation by trematode infection, urate deposition, epithelial desquamation, and hyperplasia. On their own or combined, these changes can be life-threatening or may result in broader systemic issues such as electrolytic and metabolic disturbances, which may not be detectable by routine histopathology. Furthermore, starvation, in which hypovitaminosis and dehydration coexist, may be another relevant component for the development of the reported renal lesions. Dehydration may rapidly occur in a chronically debilitated and starved marine bird with renal trematodiasis and a partially obstructed urinary tract. Dehydration by itself can induce hypovolemic shock, renal gout, and ultimately death. More studies are needed to understand the pathogenesis of nephropathy in chronically debilitated marine birds with *Renicola* sp. parasitism.

The mass mortality of tufted puffins (*Fratercula cirrhata*) and Atlantic puffins has been attributed to starvation [2,8,9]. Our mortality event shares some similarities, such as fat and muscle atrophy, but differs in the high presence of renal lesions. We hypothesize that renal lesions were exacerbated in these Atlantic puffins due to malnutrition [25]. It should be noted that the mortality event coincided with the presence of a series of winter storms extending from the North Atlantic to Spain (storms Efraín, Fein, and Gérard, from December 2022 to January 2023), which could have displaced the animals far from their feeding areas [26]. The lesions described in the skeletal muscle and the relatively high prevalence of trauma would be expected in animals suffering from exertion after struggling with bad weather conditions [27].

We frequently noticed the presence of microplastics in the ventriculus of the stranded puffins during this event. This was considered an incidental finding not related to the cause of death; however, it emphasizes the high exposure of plastic for Atlantic marine species, as evidenced in other studies [28,29]. The consequences of chronic exposure to microplastics are still unknown, although absorption and translocation of ingested plastic particles have been experimentally demonstrated [30].

## 5. Conclusions

We found that the most likely cause of death in studied Atlantic puffins stranded in winter in the Canary Islands was a combination of starvation and nephropathy triggered by bad weather events. The severity of the lesions and the high incidence of parasitism indicate that *Renicola sloanei* can cause disease and contribute to the mortality of wintering Atlantic puffins. However, starvation associated with a climatic event could be a major predisposing cause for the unusual mortality event of Atlantic puffins in the Canary Islands.

## Figures and Tables

**Figure 1 animals-15-01281-f001:**
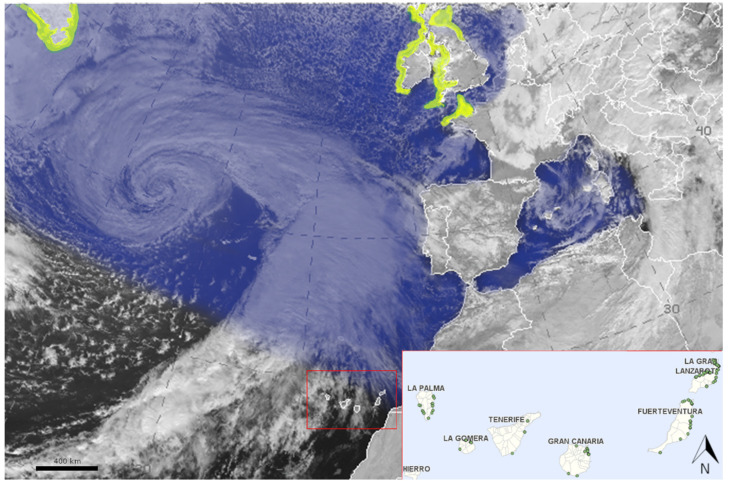
Satellite view of the storm Efraín reaching the coast of the Iberian Peninsula and Canary Islands (red square, inset). The wintering area is represented in blue, and the yellow represents the breading areas of the Atlantic puffins (image created from combined information of Meteosat satelite and DataZone by Birdlife). Inset: Strandings of Atlantic puffins in the Canary Islands. Each point represents an independent wreck event, which involved several individuals (see Appendix A for detailed information).

**Figure 2 animals-15-01281-f002:**
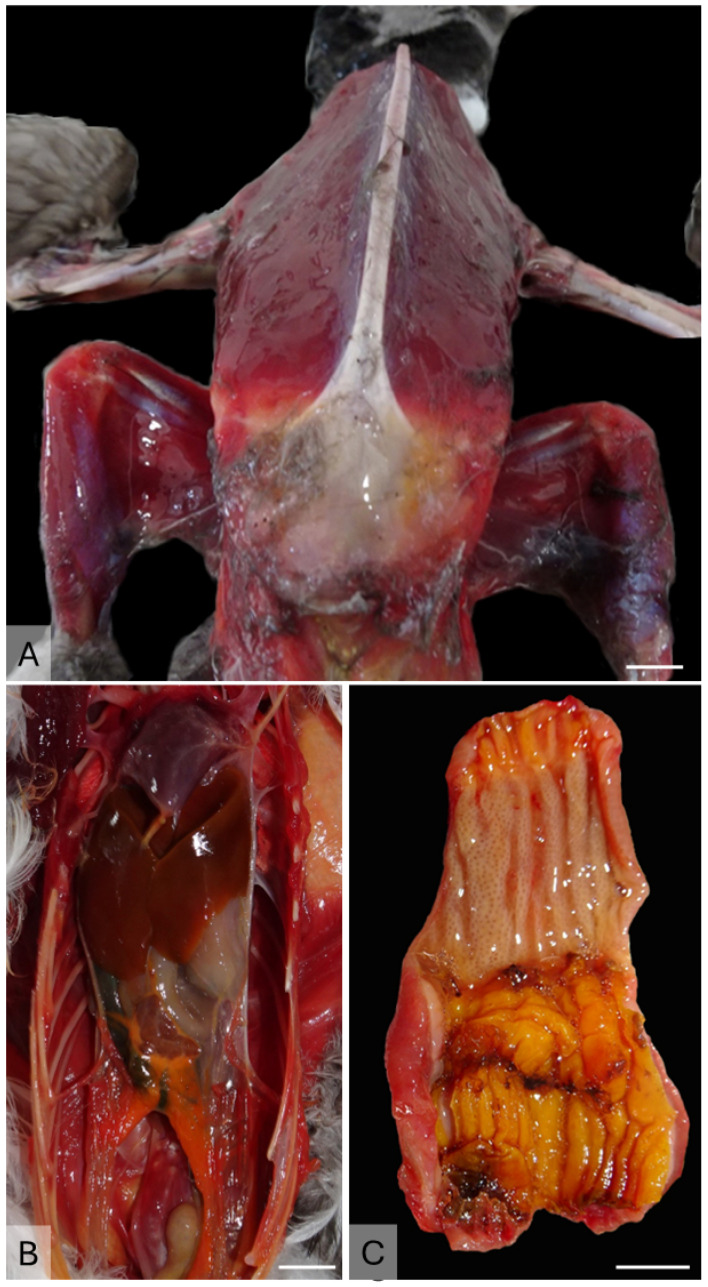
Main gross findings. (**A**) Case 46—Diffuse muscle atrophy in a cachectic Atlantic puffin (scale 1 cm). (**B**) Case 71—Celomic cavity. Depletion of visceral fat (scale 1 cm). (**C**) Case 73—Proventriculus and ventriculus. Complete absence of ingesta (scale 1 cm).

**Figure 3 animals-15-01281-f003:**
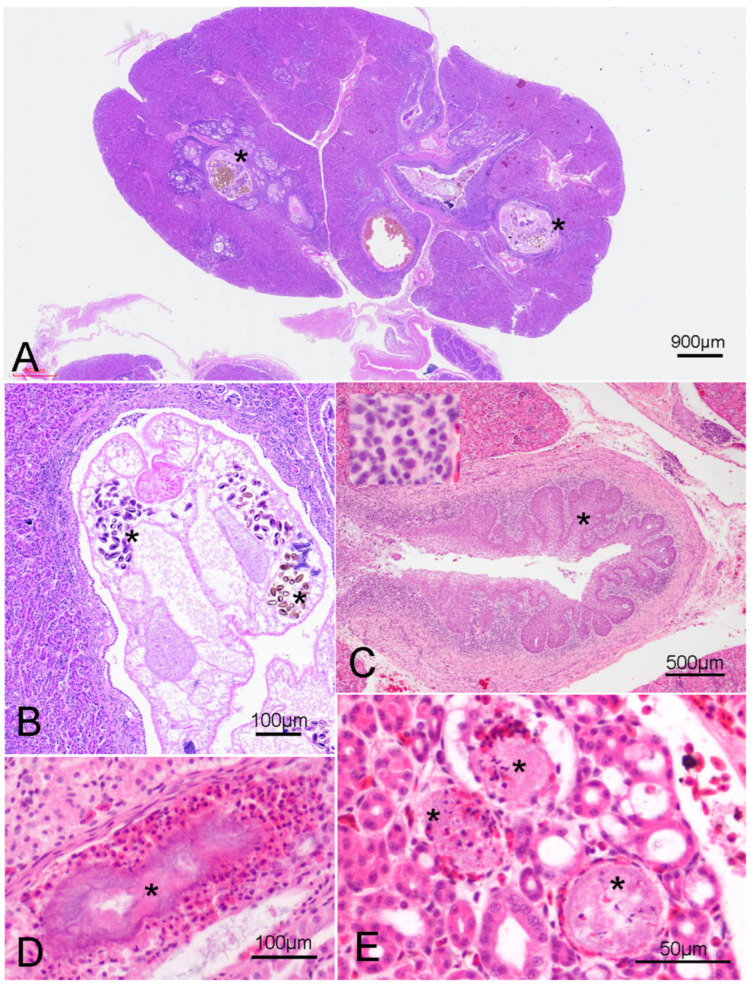
Main lesions observed in the urinary system (H&E). (**A**) Case 5—The primary ureteral branches and collecting ducts are severely dilated with intraluminal trematodes (asterisk). (**B**) Case 5—Cranial portion of a mature trematode in the lumen of a primary ureteral branch, showing numerous intrauterine ellipsoid eggs (asterisk). (**C**) Case 24—Ureteral branch. Diffuse lymphoplasmacytic ureteritis with squamous metaplasia (asterisk). Inset: Higher magnification of the lymphoplasmacytic inflammation. (**D**) Case 4—Urate crystal with heterophilic inflammation in the lumen of a collecting duct (asterisk). (**E**) Case 21—Multifocal necrosis of the epithelium of the collecting ducts (asterisk), with heterophilic inflammation.

## Data Availability

The data presented in this study are available on request from the corresponding author due to legal reasons and privacy regulation.

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
