# Peer review of "Unusual Mass Mortality of Atlantic Puffins (Fratercula arctica) in the Canary Islands Associated with Adverse Weather Events"

_animals, 2025, doi:10.3390/ani15091281_

Round 1
Reviewer 1 Report
Comments and Suggestions for Authors
Review
The data presented in this manuscript are of interest to zoologists and ecologists, as the study touches on the causes of the mass death of puffins. This is a colonial nesting species, the number of colonies of which has been declining in recent years. Finding out the reasons for this phenomenon is an important task.
I support this study, which focuses on the morphological and physiological analysis of dead birds. The sample size is quite large and representative and allows for an adequate description of the distribution of the identified pathologies, highlighting the main causes of their occurrence. One of the significant causes of mass death of birds, in particular, avian influenza H5N1, is excluded. This allowed the authors to focus on an important cause that provokes mass death of seabirds, their starvation during extreme weather events. There are interesting facts about microplastics in stomachs, infection of birds with trematodes.
I support the publication of this manuscript in Aniamals journal after revision. I have some comments on this manuscript.
- Title. In my opinion, the title needs to be clarified. The manuscript is about an extreme weather event or weather adverse event. This phrase would be more accurate than climatic adverse event.
- Simple Summary: there is no indication of the authors' main conclusion that the cause of the mass death of birds is starvation.
- Remove " climate change " from the keywords. The authors do not write about this anywhere. Instead, add "extreme weather event" or ”weather adverse event”.
- The manuscript lacks a detailed description of the extreme weather event that led to the death of the birds. The extreme event needs to be described in more detail: its duration, the storm, the inability to feed, etc. What was it?
- As is known, global warming is accompanied by an increase in the frequency of extreme weather events, and, accordingly, consequences for animals. It is advisable to mention this with references in the Introduction section.
- The manuscript discusses starvation as a cause of death of birds. But there is no information at all about the diet and hunting methods of puffins. It is necessary to add this information briefly. This will help the authors to more reasonably discuss starvation as a cause of death of puffins. This is a fish-eating species, starvation can be explained by the physical inability to obtain the main food resource due to a storm or its lack in the habitat.
- In the “Material and Methods” section, you need to add the coordinates of the place of mass death of birds or a map of the islands, on which indicate the points where the dead birds were found.
- Could some of the puffins that were found to have mechanical damage have died as a result of storm events? This point is not discussed, although the results contain information about injuries to 13 birds. These are usually young birds. What is the age distribution of the injured birds?
- What is the age and sex composition of birds infected with trematodes? This is also important information and can be discussed.
Technical remarks
There is a repetition of lines: lines 111-112 are duplicated by lines 118-119
Author Response
Dear reviewer, many thanks for your suggestions and your comments. We really appreciated all of them, as we think they have contributed to the improvement of this work. Following, we response point by point to your comments. The recommended changes has been implemented accordingly throughout the text. Also, major changes has been performed in the supplementary material to accomplish reviewers suggestions.
We hope, you find our article suitable for publication in the journal Animals.
Comment 1. Title. In my opinion, the title needs to be clarified. The manuscript is about an extreme weather event or weather adverse event. This phrase would be more accurate than climatic adverse event.
The title has been changed accordingly.
Comment 2. Simple Summary: there is no indication of the authors' main conclusion that the cause of the mass death of birds is starvation. Remove " climate change " from the keywords. The authors do not write about this anywhere. Instead, add "extreme weather event" or ”weather adverse event”.
We implemented the suggested modification accordingly.
Comment 3. The manuscript lacks a detailed description of the extreme weather event that led to the death of the birds. The extreme event needs to be described in more detail: its duration, the storm, the inability to feed, etc. What was it?
Dear reviewer, many thanks for this comment. We agree with you, and he have incorporated a new paragraph in the introduction for the description of this series of winter storms:
“A series of winter storms battered the coasts of the Iberian Peninsula, France, and the United Kingdom between December 2022 and January 2023. These storms were named by various meteorological organizations as storms Efraín, Fien, and Gérard. In particular, the formation of the storm Fien was rapidly followed by the storm Gérard. They were characterized by the existence of a very powerful jet stream (exceeding 300 km/h) at high atmospheric levels, which crossed the entire North Atlantic from the Labrador Peninsula to the Iberian Peninsula (see detailed meteorological information in https://www.aemet.es/es/conocermas/borrascas/2022-2023). Extreme weather conditions may represent a threat-life event even for a well-adapted seabird. As global warming is accompanied by an increase in the frequency of extreme weather events, this may represent a new scenario for the Atlantic puffins that get lost from their migratory route.”
Comment 4. As is known, global warming is accompanied by an increase in the frequency of extreme weather events, and, accordingly, consequences for animals. It is advisable to mention this with references in the Introduction section.
This has been addressed accordingly with the new paragraph of the introduction.
Comment 5. The manuscript discusses starvation as a cause of death of birds. But there is no information at all about the diet and hunting methods of puffins. It is necessary to add this information briefly. This will help the authors to more reasonably discuss starvation as a cause of death of puffins. This is a fish-eating species, starvation can be explained by the physical inability to obtain the main food resource due to a storm or its lack in the habitat.
A new phrase has been added in the beginning the introduction:
“The Atlantic puffin (Fratercula arctica) is a seabirdcharadriiform bird renowned for its exceptional diving capabilities and transoceanic migratory behaviour. Phylogenetic analyses have identified several subspecies that nest in various geographical locations, including northern Europe, Greenland, and North America [1, 2, 3]. Their diets are rich in small fishes, which may vary according differeng populations, but frequently include sand eels (Hyperoplus spp.), herrings (Clupea sp.), and capelin (Mallotus sp.) [4].
Comment 6. In the “Material and Methods” section, you need to add the coordinates of the place of mass death of birds or a map of the islands, on which indicate the points where the dead birds were found.
The figure 1 has been changed to accommodate to the reviewer’s suggestion. Now the figure 1 illustrates the location of the different stranding events in the archipelago.
Comment 7. Could some of the puffins that were found to have mechanical damage have died as a result of storm events? This point is not discussed, although the results contain information about injuries to 13 birds. These are usually young birds. What is the age distribution of the injured birds?
First, we would like to emphasize that all the birds in which a diagnosis of “trauma” was realized were also emaciated. In these cases, we stablish the trauma as a secondary mortality factor. We normally found trauma as the related cause of dead in emaciated individuals, but we normally consider the malnutrition (such our case) as the primary mortality factor. We consider this aspect very technical/forensic (primary vs. secondary mortality factor as the cause of death), and we think we should not enlarge too much the discussion. However, we can discuss it if the reviewer considers it essential for the comprehension of the results.
We already discuss (very briefly) the role of the trauma in our cases in the lines 299-301.
“The lesions described in the skeletal muscle and the relatively high prevalence of trauma would be expected in animals suffering from exertion after struggling with bad weather conditions”
We have checked the bird with diagnosis of trauma, and we found in 13 traumatized birds, 2 juveniles and 6 adults (in 5 individuals the age could not be determined). Please take in consideration that our methodology is focused in the diagnosis of the cause of dead, and we would like to keep this as the main topic of the article. Although we take some basic biological data during the necropsy, we usually don’t attempt to determinate the age of the birds as precisely as in other scientific areas (i.e. biology, ornithology). We apologize because potentially our biological determinations are very limited and based on the morphology of the gonads (i.e. mature vs. immature). We would like to report a simple but in somehow relevant pathological report of what we observed in this episode.
Comment 8. What is the age and sex composition of birds infected with trematodes? This is also important information and can be discussed.
Thanks for that recommendation. We will have it in great consideration. We are already preparing another work more centred in the epidemiological data of the parasitism.
We can analyse and interpretate what you ask, however, we believe that the interpretation would not reflect the true, as we think we are underestimate the infection rate base on histopathological result. This is already discussed in the text:
“Careful investigation of the ureters during gross examination, and the inclusion of representative samples of different renal divisions, as well as ureters for histopathology, may increase the chance of detecting the trematode. However, molecular analysis with the amplification of the NADH dehydrogenase subunit 1 gene may be more sensitive for detecting the parasite in renal tissue. Clarification of the best diagnostic method for this condition merits further investigation.”
We are planning to use molecular analysis after validation of the diagnosis tool (non-invasive cloacal sample). The validation of a more sensitive and sensible diagnosis tool will be better to know if the prevalence is different in males and females, or in young rather than adults.
However, in the supplementary material, we have included the information of the infection rates that we have determined during histopathology, organized in males and females, as well as in immature and mature birds.
Additionally, and as requested for the other reviewer, we tested chi square to test the hypothesis if the sex or the maturity affect for the parasitism, and we did not find significative differences in the rate of parasitism in different sex or ages.
Comment 9. Technical remarks
There is a repetition of lines: lines 111-112 are duplicated by lines 118-119
Thanks. The issue has been addressed.

Reviewer 2 Report
Comments and Suggestions for Authors
The study describes the anatomopathological findings of an unusual mortality event of Atlantic puffins observed during the non-breeding period in the Canary Islands.
The postmortem investigation revealed that examined individuals were severely emaciated and suffered from nephropathy. The Author suggest that this mortality event was a consequence of starvation associated with bad weather conditions during a migratory movement. Avian influenza was ruled out as the cause of death in the examined individuals.
The study is interesting in the context of scarcity of data about the causes of massive mortality events of stranded seabirds
The study is generally well designed and performed. However, I have some comments and corrections provided in the attached pdf file and below:
Simply Summary: please simplify language – I provided some suggestions in the attached pdf file
Abstract: provide sample size
Materials and methods:
- provide details about measurements including tools – ruler, caliper etc.
- It would be good to use some at least basic statistical analyses in the study – for example you may compare the proportion of various BS between adults and immatures, females and males using chi-square or G test.
Results
Lines 123-124 and also part of Discussion – body mass during the non-breeding period can be different from the breeding period. So data reported here should be compared with non-breeding data; Look for data from non-breeding period e.g. here: Anker-Nilssen, T., Jensen, J. K., & Harris, M. P. (2018). Fit is fat: winter body mass of Atlantic Puffins Fratercula arctica. Bird Study, 65(4), 451–457. https://doi.org/10.1080/00063657.2018.1524452
Anyway, this thread better fits to Discussion than to Results
Figure 1 – if you want to show normal distribution it would be better to show density plot instead of histogram with body mass categories
As I mentioned before - if the sex and age were identified I would expect to compare body mass between these group.
It would be also good to compare body mass between individuals in different state (e.g. in relation to BS categories). I guess that it is not possible to all age-sex categories but some categories can be combined to have considerable same size for the compared groups.
It is written „Necropsies were performed on 147 animals” but later is written “Both juveniles (n=35) and adults (n=67) were present”. What about age of lacking 45 individuals (147 vs 102)? Were they unidentifiable?
Figure 3 – consider adding arrows or/and squares to highlight described features – the are not obvious for non-specialists in parasitology. provide linear scale
Climatic events was blamed as the cause of massive death of the examined puffins stranded in the Canary Island. This climatic event should be described in details as seabirds are generally adapted to changes in environmental conditions including strong storms. “A climatic adverse event„ is mentioned in the title and Conclusions n but it is only mentioned in one sentence in Discussion. Maybe it should be called less enigmatically as “winter storm”?
When the trematode could have infested the examined Puffins? I would suggest to describe shortly the life cycle of this parasite. During the past breeding season? If so it can be interpreted as a carry-over effect of breeding period affecting survival during the non-breeding period (at least in the case of severe inflammatory reactions), right?

Author Response
Dear reviewer, many thanks for your suggestions and your comments. We really appreciated all of them, as we think they have contributed to the improvement of this work. Following, we response point by point to your comments. The recommended changes has been implemented accordingly throughout the text. Also, major changes has been performed in the supplementary material to accomplish reviewers suggestions.
We hope, you find our article suitable for publication in the journal Animals.
Comment 1: Results. Lines 123-124 and also part of Discussion – body mass during the non-breeding period can be different from the breeding period. So data reported here should be compared with non-breeding data; Look for data from non-breeding period e.g. here: Anker-Nilssen, T., Jensen, J. K., & Harris, M. P. (2018). Fit is fat: winter body mass of Atlantic Puffins Fratercula arctica. Bird Study, 65(4), 451–457. https://doi.org/10.1080/00063657.2018.1524452. Anyway, this thread better fits to Discussion than to Results
These changes were implemented accordingly in the discussion.
Comment 2: Figure 1 – if you want to show normal distribution it would be better to show density plot instead of histogram with body mass categories. As I mentioned before - if the sex and age were identified I would expect to compare body mass between these group. It would be also good to compare body mass between individuals in different state (e.g. in relation to BS categories). I guess that it is not possible to all age-sex categories but some categories can be combined to have considerable same size for the compared groups.
Dear reviewer. Your suggestion is very appreciated, and we would really like to perform more investigation about the weight and the body condition of these, emaciated animals. But we feel that would be beyond the current work, who pretend to describe the pathological findings in an episode of mass mortality.
We think the investigation by different multidisciplinary approaches (which may include systematic statistical approach, metabolic research, tissular weighing) would be very beneficial for the demonstration of the catabolic state of this animal. We have already a quite lower mean body weight that it is recorded for the species, and the reproductive period. We also provide morphological evidence that the animals were emaciated. However, we agree that the body mass data may be well appreciated for many scientists interested in this area. For that reason, we have incorporated this information in the form a new supplementary table. It has been added to “Supplementary material”.
We hope this would partially fill the gap between the morphology and the numeric data.
Comment 3: It is written „Necropsies were performed on 147 animals” but later is written “Both juveniles (n=35) and adults (n=67) were present”. What about age of lacking 45 individuals (147 vs 102)? Were they unidentifiable?
Sorry about the numbers. We have improved the case selection and only animals with complete standardized necropsies, has been included in the Necropsy results. We justify this change for better clarification of the results, as in this group the sex and the degree of maturity is always known. We have changed the sentences for clarification:
The registered mean body mass of 136 individuals was 256.0±47.4 g. The individual data can be found in the supplementary table (biometry) . Complete standardized necropsy was performed on 81 animals. there were 54 females (66.66%) and 27 males (33.34%). Both juveniles (n=52) and adults (n=29) were present.
The most consistent gross findings were generalized muscle atrophy (Figure 2A) and serous fat atrophy (Figure 2B). These conditions were evident in 96.3% (78/81) of the individuals for which body score (BS) could be determined. The BS ranged from cachectic (BS1, n=40) to thin (BS2, n=38), with only 2,5% of individuals (n=2) having a BS of 3. In all animals, the ventriculus and proventriculus were empty (Figure 2C), except for the frequent presence of microplastics (i.e., fragments of plastic less than 5 mm) in the ventriculus.
Comment 4: Figure 3 – consider adding arrows or/and squares to highlight described features – the are not obvious for non-specialists in parasitology. provide linear scale
Scales have been added in histological panels with helminths. Also, asterisks have been placed in the figure for better orientation.
Comment 5: Climatic events was blamed as the cause of massive death of the examined puffins stranded in the Canary Island. This climatic event should be described in details as seabirds are generally adapted to changes in environmental conditions including strong storms. “A climatic adverse event„ is mentioned in the title and Conclusions n but it is only mentioned in one sentence in Discussion. Maybe it should be called less enigmatically as “winter storm”?
Thanks for the comment. We have added a new paragraph in the introduction for the description of the winter storms related to the event. We included the original meteorological information that alerted and named the storms, for consultations. The new paragraph is:
“A series of winter storms battered the coasts of the Iberian Peninsula, France, and the United Kingdom between December 2022 and January 2023. These storms were named by various meteorological organizations as storms Efraín, Fien, and Gérard. In particular, the formation of the storm Fien was rapidly followed by the storm Gérard. They were characterized by the existence of a very powerful jet stream (exceeding 300 km/h) at high atmospheric levels, which crossed the entire North Atlantic from the Labrador Peninsula to the Iberian Peninsula (see detailed meteorological information in https://www.aemet.es/es/conocermas/borrascas/2022-2023). Extreme weather conditions may represent a threat-life event even for a well-adapted seabird. As global warming is accompanied by an increase in the frequency of extreme weather events, this may represent a new scenario for the Atlantic puffins that get lost from their migratory route.”
Comment 6: When the trematode could have infested the examined Puffins? I would suggest to describe shortly the life cycle of this parasite. During the past breeding season? If so it can be interpreted as a carry-over effect of breeding period affecting survival during the non-breeding period (at least in the case of severe inflammatory reactions), right?
We have added a new phrase to explain more the life cycle of the parasite in the introduction, and linked it with the diet of the Atlantic puffins, as suggested with other reviewer:
“…The metacercarial stage is the infective form for the seabird, and they are encysted in the mesenteries connecting the pyloric caeca of the host fish. They can be commonly found in European sprat (Clupea sprattus), and mediterranean sardines (Sardina pilchardus) [11] ...”
However, we think that questions merit further analysis including, PCR of prey in the non-breeding and breeding season, and full molecular characterization of the life cycle in the definitive host, in different temporalities. Based on our personal observations of histopathology of marine birds, vertical transmission would be a potential first infection, but reinfections should be considered. However, that would be beyond the objective of this work, and we think we don’t have enough evidence to argue that.
What we do defend, throughout the work, is that parasitic infestation together with starvation can be fatal. We think this is the central idea; however, we have included in the supplementary material 2 new epidemiological information including age category and sex.
Additional answers:
Previous Lines 224-225. Thanks for your suggestions. In this case, we prefer to group the moderate-to-severe category, because in some cases, moderate lesions can be as deadly as severe. That would be the case in our work. When we are able to observe the lesions histologically, that would mean a severe metabolic compromise in the homeostasis of the individual. With lesions such as uretheral obstruction (secondary to inflammation, trematode presence, urate deposition, epithelial desquamation and hyperplasia), acute epithelial damage, or necrosis, in combination, can be deadly or being the result of systemic alterations (ions alterations, plasma acidification, shock). We think we have discussed this later in the text, in a different paragraph that can be subtitled (but is not) “pathogenesis”.

Round 2
Reviewer 1 Report
Comments and Suggestions for Authors
The manuscript has improved after the corrections, the authors have made corrections to it in accordance with my comments. I recommend this version for publication.
Author Response
Thanks for your revision. Regards.
Reviewer 2 Report
Comments and Suggestions for Authors
The manuscript has been improved. The Authors addressed majority of my concerns from the previous review. The Authors did not address all my comments and suggestions from my first review. I do not expect to accept all my suggestion but I expect at least some comment.
Some points listed below still require additional attention.
- I think that it is worth to show some results with division into age or sex groups. Providing those data in supplementary file is not the same as presenting it in the main text. I do not agree with Authors that “that would be beyond the current work, who pretend to describe the pathological findings in an episode of mass mortality” as some information like age dependent exposure to infections or being stranded may be very important;
- I am glad that Authors provided a map with location of found stranded individuals. Add standard map features: linear scale and north arrow or geographical grid (latitude and longitude). It would be good to indicate with the circle size the numbers of found stranded individuals. Also consider showing additionally the larger map with Atlantic and the range of the winter storms observed between December 2022 and January 2023 being the reason of death of the examined puffins; I see from the map that you have coordinates of all stranded individuals – it would be good to provide them in supplementary materials. Currently only the name of island has been provided.
- Simple summary – I think that still you can simplify some words used there (e.g. etiopathological investigation, postmortem, concomitant infestation with a renal parasite); maybe they should be stated in more descriptive way (examination of dead individuals, they have parasites in kidney, etc.)
- Line 43 – not finished sentence about Influenza
- Line 57 – do not use “their” as you introduced puffin in a singular form as the species; similar problem in line 57 and 72
- Lines 67-75 – describe how the parasite is infecting avian host
- Lines 85 -87 – “this may represent a new scenario for the Atlantic puffins that get lost from their migratory route„ – it was not stated that Canary Islands are outside of the wintering range – it would be good to provide a map with wintering range of the Atlantic Puffin – you can ask the BirdLife International for such layers for the purpose of this study. According to their map a non-breeding range is covering some Canary Islands. See here: https://datazone.birdlife.org/species/factsheet/atlantic-puffin-fratercula-arctica; you started Discussion form the sentence “The Atlantic puffin is a rare visitor to the Canarian coast” (line 215). Provide reference to this information (especially in the context of the mentioned map)
- Line 115 – provide criteria for ageing individuals (e.g. reference)
- Figure 2 – provide linear scale in photos, Fig. 3 – provide linear scale in all photos
Author Response
Dear reviewer. Again, we must thank all your commentaries, suggestions and corrections. All of them are very welcome and we feel our work have improved considerably thanks of your revision.
Following, we have responded, point by point to all your commentaries:
Commentary 1: I think that it is worth to show some results with division into age or sex groups. Providing those data in supplementary file is not the same as presenting it in the main text. I do not agree with Authors that “that would be beyond the current work, who pretend to describe the pathological findings in an episode of mass mortality” as some information like age dependent exposure to infections or being stranded may be very important;
Thanks for the appreciation and the suggestion. We also believe the infection rates in young and adults, as well as differences by sex is very relevant for the diagnostic, biological, parasitological, and ecological point of view. We did not pretend to overview your previous suggestion, only to justify the limitation of our diagnostic methods (already discussed in the text).
However, we have added the information about the incidence of infection in both sexes, as well as in immature, and mature animals. This has been added in the Histopathological results.
Also, we added a new phrase in the discussion, to highlight the data of parasitic incidence obtained in the results:
“Although our histopathological analysis revealed some early data of parasitic incidence of Renicola sloanei in the Atlantic puffin, this diagnostic method may underestimate the actual infestation rate”.
We hope with these modifications we accomplish your kind suggestion.
Commentary 2: I am glad that Authors provided a map with location of found stranded individuals. Add standard map features: linear scale and north arrow or geographical grid (latitude and longitude). It would be good to indicate with the circle size the numbers of found stranded individuals. Also consider showing additionally the larger map with Atlantic and the range of the winter storms observed between December 2022 and January 2023 being the reason of death of the examined puffins; I see from the map that you have coordinates of all stranded individuals – it would be good to provide them in supplementary materials. Currently only the name of island has been provided.
Many thanks for these recommendations. Please note that now figure 1 represents a larger area, in which the storm is visualized, and the wintering and breeding area of the Atlantic puffins have been represented. Standard map features (i.e. geographical grid, North arrow, and approximate distance scale) has been added.
Also, we have included in the supplementary material the coordinates (UTM based).
We opted to not include the circle with the numbers of individuals as it may impaired the correct visualization of the stranding points in the inset. However, we included a new section in the material and methods and results, to better define the Stranding events, separately from the necropsies. We also included a new diagram in the supplementary material to illustrate the stranding events.
Commentary 3: Simple summary – I think that still you can simplify some words used there (e.g. etiopathological investigation, postmortem, concomitant infestation with a renal parasite); maybe they should be stated in more descriptive way (examination of dead individuals, they have parasites in kidney, etc.)
We have simplified more the Simple summary.
Commentary 4: Line 43 – not finished sentence about Influenza
Sorry for that. It is amended accordingly.
Commentary 5: Line 57 – do not use “their” as you introduced puffin in a singular form as the species; similar problem in line 57 and 72
Sorry for that. It is amended accordingly.
Commentary 6: Lines 67-75 – describe how the parasite is infecting avian host
A new phrase has been added to describe the process of infection for the bird. It can be found in the third paragraph of the introduction:
The seabird usually gets the parasite through the ingestion of infected fish, After the excystation of the metacercaria in the bird gastrointestinal tract, the parasites migrate to reach it definitive location at the bird urinary tract [11].
Commentary 7: Lines 85 -87 – “this may represent a new scenario for the Atlantic puffins that get lost from their migratory route„ – it was not stated that Canary Islands are outside of the wintering range – it would be good to provide a map with wintering range of the Atlantic Puffin – you can ask the BirdLife International for such layers for the purpose of this study. According to their map a non-breeding range is covering some Canary Islands. See here: https://datazone.birdlife.org/species/factsheet/atlantic-puffin-fratercula-arctica; you started Discussion form the sentence “The Atlantic puffin is a rare visitor to the Canarian coast” (line 215). Provide reference to this information (especially in the context of the mentioned map)
Dear reviewer, we have added a new phrase in the beginning of the discussion, and we added the reference supported by the author (see reference 18). We also support new figure 1 with the wintering range of the Atlantic puffins, for better illustration of the results.
We hope this would reflect accordingly your suggestions.
Commentary 8: Line 115 – provide criteria for ageing individuals (e.g. reference)
A new reference has been added for determining maturity in avian species.
Commentary 9: Figure 2 – provide linear scale in photos, Fig. 3 – provide linear scale in all photos
Dear author, please find linear scale in all the photos of the figure 2 and figure 3.